# Rapid molecular testing or chest X-ray or tuberculin skin testing for household contact assessment of tuberculosis infection: A cluster-randomized trial

Menonli Adjobimey[1,2], Anete Trajman[3,4,5], Mayara Lisboa Bastos[5,6], Chantal Valiquette[5], Diana Gibson[5], Frimege Djohoun[2], Olivia Oxlade[5], Federica Fregonese[5], Dissou Affolabi[2,7], Valentin Kouchade[2], Elisa Aguiar[8], Renata Spener-Gomes[9,10,11], Marcelo Cordeiro-Santos[9,10,12], Renato T. Stein[13,14], Marcelo Scotta[13,14], Andrea Benedetti[4,5], Dick Menzies[4,5]*

**1** Department of Public Health, Faculty of Health Sciences, University of Abomey-Calavi, Cotonou, Benin, **2** Respiratory Service, National Tuberculosis Program, Cotonou, Benin, **3** Internal Medicine Department, Universidade Federal do Rio de Janeiro, Rio de Janeiro, Brazil, **4** Faculty of Medicine, McGill University, Montreal, Canada, **5** McGill International TB Centre, Research Institute of the McGill University Health Center, Montreal, McGill University, Montreal Canada, **6** Family Medicine, Max Rady College of Medicine, University of Manitoba, Winnipeg, Canada, **7** Department of Biology, Faculty of Health Sciences, University of Abomey-Calavi, Cotonou, Benin, **8** Department of Pediatrics, Universidade Federal do Rio de Janeiro, Rio de Janeiro, Brazil, **9** Gerência de Micobacteriologia, Fundação de Medicina Tropical Dr. Heitor Vieira Dourado, Manaus, Brazil, **10** Escola de Medicina, Universidade do Estado do Amazonas, Manaus, Brazil, **11** Departamento de Saúde Coletiva, Universidade Federal do Amazonas, Manaus, Brazil, **12** Escola de Medicina, Universidade Nilton Lins, Manaus, Brazil, **13** Programa PROADI-SUS-TBPed, Hospital Moinhos de Vento, Porto Alegre, Brazil, **14** Department of Pediatrics, Pontifícia Universidade Católica, Porto Alegre, Brazil

\* dick.menzies@mcgill.ca

## Abstract

### Background

The World Health Organization recommends evaluation of all household contacts (HHC) of index tuberculosis (TB) patients for TB disease (TBD) and TB infection (TBI). Tests to identify TBI and TBD are preferred but can be skipped in persons living with HIV and children <5 years. There is equipoise on the need for these tests in other HHC.

### Methods

We conducted a superiority, open label cluster-randomized trial in Benin and Brazil to compare three strategies to evaluate HHC aged 5–50 of persons newly diagnosed with drug susceptible pulmonary TBD: Standard: tuberculin skin testing (TST) for TBI and if positive, chest X-ray (CXR) to rule out TBD; rapid molecular test (RMT): same as Standard, except CXR replaced by an RMT; and No-TST: CXR for all but no TST. Randomization was computer-generated and stratified by country, in blocks

which permits unrestricted use, distribution, and reproduction in any medium, provided the original author and source are credited.

**Data availability statement:** The data from this study will be available from a public repository. https://doi.org/10.25934/PR00011500.

**Funding:** The study was supported by operating grants from the Canadian Institutes of Health Research (FDN-143350 to DM) and the Brazilian Ministry of Health (parecer técnico 3/2021, NUP 25000.012788/2021-61 to AT). Study authors were also supported by salary awards from Canada Research Chair award to DM, Conselho Nacional de Desenvolvimento Científico e Tecnológico of Brazil to AT, and MCS. The funders had no role in the study design, data collection and analysis, decision to publish, or preparation of the manuscript.

**Competing interests:** The authors have declared that no competing interests exist.

**Abbreviations:** CXR, chest X-ray; GEE, generalized estimating equations; HHC, household contacts; IGRA, interferon gamma release assays; PEPFAR, President's Emergency Plan for AIDS Relief; PLHIV, people living with the human immunodeficiency virus; RMT, rapid molecular test; TB, tuberculosis; TBD, TB disease; TBI, TB infection; TPT, TB preventive therapy; TST, tuberculin skin testing; WHO, World Health Organization.

of variable length. The primary outcome was TB preventive therapy (TPT) initiation among HHC considered eligible (positive TST, if done, and no evidence of TBD on CXR or RMT). Secondary outcomes were: completion of investigations to detect TBI and TBD, detection of TBD, TPT completion, severe adverse events, and societal costs.

## Results

Among 1,589 participating HHC enrolled from 29 January 2020, to 30 November 2022, 474 were randomized to the standard, 583 to the RMT, and 532 to the no-TST strategies; all were included in the analyses. Of 848 HHC considered eligible for TPT, 802 (94.6%) initiated TPT, with no difference between strategies (95%, 94%, and 95% for the standard, RMT, and no-TST strategies, respectively). Of the secondary outcomes, protocol-mandated investigations to detect TBI and exclude possible TBD were completed for 93.4% overall, with slight differences between arms (93%, 95%, and 93% for the standard, RMT, and no-TST strategies, respectively). Adverse events resulting in discontinuation of TPT occurred in 3 (0.4%) participants in total (with 1, 0, and 2 events among participants in the Standard, RMT, and no-TST arms, respectively). The proportion completing TPT was similar with Standard and RMT strategies but was 13% lower (95% confidence interval: 3% to 23% lower) with the No-TST strategy. Societal costs per HHC completing investigations were $61 ($56–$65) with the standard strategy, compared to $52 ($49–$55) with the RMT strategy and $74 ($72–$77) with the no-TST strategy.

## Conclusion

This randomized trial provides ~~high-quality~~ evidence that TST followed by selected use of CXR or an RMT to exclude disease can achieve high rates of TPT initiation at reasonable costs. A limitation of the trial is the potential study effect, which may have affected adherence by providers and HHCs. RMT could replace CXR in the management of HHC in resource limited settings.

## Registration

clinicaltrials.gov NCT04528823

---

### Author summary

There are many potential barriers in the investigation of household contacts (HHC) of persons with TB disease. These barriers often result in HHC not starting TB preventive treatment (TPT), which is recommended by the World Health Organization (WHO), because HHC are at high risk of developing TB disease themselves. We conducted an open-label cluster randomized trial in Benin and Brazil to compare three strategies of investigation and management of HHC. The three strategies were the **standard** strategy recommended by WHO, an

experimental strategy using a rapid molecular test (**RMT**) instead of chest X-ray (CXR), and an experimental strategy of CXR performed in all HHC, but no Tuberculin skin testing (**no-TST**). We enrolled 1,589 HHC, of whom 474 were randomized to the standard arm, 583 to the RMT arm, and 532 to the no-TST arm. Among the HHC who were considered to have TB infection, but not disease more than 95% started TPT, of whom only 68% completed this treatment. The proportion who started TPT was the same in three strategies, although completion was lower in the no-TST strategy. Only three persons stopped therapy because of side effects of the medications. Overall costs for the strategies were lowest with the RMT, and highest with the noTST. We conclude that more intense investigations can identify people at higher risk for TB disease, and do not pose a barrier if services are well organized. Rapid molecular tests appear promising to replace CXR in the management of HHC, and may resolve a long-standing bottleneck in many countries.

## Introduction

In 2023, globally tuberculosis disease (TBD) developed in 10.8 million persons and was responsible for 1.25 million deaths [1]. Prevention of TBD through treatment of TB infection (TBI) is a key strategy for TB elimination [2] as nearly one quarter of the world's population is estimated to have TBI [3], of whom approximately 10% will develop the disease, maintaining the transmission chain. For these reasons, the United Nations High-Level Meeting (UN-HLM) held in September 2023 committed to provide TB preventive treatment (TPT) to 45 million people by 2027 [4]. TPT regimens include 6 or 9 months monotherapy with Isoniazid, 4 months monotherapy with rifampin or 3 months Isoniazid combined with Rifampin or Rifapentine [5]. Five-year targets for TPT provision, set in 2018, were exceeded for people living with the human immunodeficiency virus (PLHIV). A major reason for this achievement was that provision of TPT to PLHIV became a major target for programs funded by the President's Emergency Plan for AIDS Relief (PEPFAR). After 2018 PEPFAR allocated substantial resources to provision of TPT, with the results that millions of PLHIV received TPT between 2018 and 2023 [6]. However, TPT provision fell far short of the UN targets for young children and all other household contacts (HHC) [1].

There are many potential barriers to testing for TBI and ruling out TBD, that together may result in substantial reduction of persons starting TPT [7]. Recognizing this, the World Health Organization (WHO) has recommended that if tuberculin skin testing (TST), or interferon gamma release assays (IGRA) or chest X-ray (CXR) are not available, TPT should be given to all those at highest risk after clinical evaluation [5]. Performance of TST/IGRA is preferred where feasible [5], given that TPT is burdensome for patients, and may cause serious adverse events, including fatal hepatotoxicity [8]. The great majority of evidence from systematic reviews and clinical trials has shown that TST distinguishes those who will benefit from TPT from those who will not, even in immunocompromised populations, such as PLHIV [9–15]. Systematic reviews of observational studies in low- and middle-income countries have estimated that approximately 50% of HHC [16], and 69% of PLHIV [17] are TST-negative. Hence, without TBI testing, the majority of persons who would receive TPT would be exposed to potential harms, without significant benefit.

Before initiating TPT it is also important to exclude the possibility of TB disease to avoid inadvertent monotherapy which could result in acquired drug resistance—an important potential harm. Systematic reviews [18,19] have estimated that symptom screening has 80% sensitivity for detection of TBD in untreated PLHIV, but sensitivity is less than 50% in PLHIV receiving antiretroviral treatment [20], and 27.7% in the general population [21]. Although systematic reviews of randomized trials have concluded that TPT does not increase the risk of resistance to rifampicin [22], or isoniazid [23], in all trials included in these reviews, participants underwent symptom screen and CXR to exclude active TBD before initiating TPT.

We conducted a cluster randomized trial in Benin and Brazil to compare the yield and costs of two experimental evaluation strategies with the standard strategy recommended by WHO, among HIV uninfected HHC aged 5 and older. One experimental strategy replaced CXR with a rapid molecular test (RMT), and the other experimental strategy did not use TST, but CXR was performed in all HHC.

## Methods

### Study design

As described elsewhere [24], we conducted a superiority, open-label cluster-randomized multicenter trial with three arms of equal size. Clusters were defined as all members of the same household of patients with newly diagnosed pulmonary TBD. We randomized the first eligible HHC who provided signed informed consent to participate to one of the three strategies. We assigned all subsequently enrolled members of the same household to the same arm. Randomization was computer-generated and stratified by country, in blocks of variable length.

### Study population: Eligibility, inclusion, and exclusion criteria

**Study sites.** At eligible clinics in Porto-Novo and Cotonou, Benin, and Rio de Janeiro, Porto Alegre, and Manaus, Brazil, more than 80 TB patients were diagnosed and treated annually. We selected these clinics as they were representative of the diagnostic facilities available in these countries. In Benin, the study was conducted at two referral clinics specialized in TB. In Brazil, study sites included out-patient facilities at tertiary centers, as well as primary care community clinics.

**Index TB patients.** Eligible persons were aged 13 or older in Benin and 14 or older in Brazil, diagnosed to have pulmonary TB disease within the past 30 days on the basis of a positive GeneXpert MTB/Rif Ultra (GX). If GX detected rifampin resistance, then culture with drug-susceptibility test for *Mycobacterium tuberculosis* was performed in both countries; otherwise, cultures were not routinely done for index TB patients. We excluded index patients if they had no eligible HHC or had confirmed rifampicin or multiple-drug TB resistance. We approached eligible persons for permission to invite their HHC to participate.

**Household contacts.** Eligible HHC were aged 5–50 years old, not known to have had a CXR or TBI test in the past 3 months. We excluded children under 5 and PHIV because of different WHO guidance for their management following TB exposure, particularly expedited initiation of TPT [5]. For HHC with unknown HIV status, voluntary HIV testing was offered according to National Guidelines [20,21]. Persons over the age of 50 were excluded because of the higher risk for severe adverse events (especially hepatotoxicity), a particular consideration for the No-TST Strategy. In Benin, following NTP guidelines, pregnant women were included after the first trimester and underwent CXR, when indicated, with appropriate shielding. In Brazil, HHC who were pregnant were excluded.

### *Interventions* (see S1 Fig)

**Standard.** This strategy is based on the current WHO recommended algorithm for HHC that are HIV-negative and aged ≥5 years [25,26]. HHC underwent initial symptom screening and TST. (IGRA's were allowed but were not available at participating facilities). At the time of TST reading (i.e., after 48–72 h), patients with symptoms at the time of initial screening were reassessed. If the TST was positive (≥5 mm in Brazil, or ≥10 mm in Benin), or persistent cough, sputum, fever, or anorexia/weight loss were reported, participants underwent a CXR. If CXR was judged possible TBD by the radiologist, one to three spontaneous sputum samples were sent for GeneXpert MTb-RIF Ultra. Once TBD was excluded, TPT was recommended.

**Experimental 1: Rapid molecular tests (RMTs).** The key difference from Strategy 1 is that an RMT replaced CXR. In both countries CXR was a recurrent bottleneck in the investigation of HHC due to equipment breakdown, lack of reagents or other supplies, and lack of trained radiologist for timely CXR interpretation. In this study the RMT performed in all facilities was the GeneXpert MTB/RIF Ultra (Cepheid). HHC underwent initial symptom screen and TST. If TST was positive or there were persistent symptoms (as defined for Standard Strategy), participants provided a spontaneous sputum sample for RMT. Once TBD was excluded, TPT was recommended.

**Experimental 2: No-TST.** All participants underwent symptom screen and CXR. Participants without symptoms and a normal CXR were offered TPT immediately. This meant that the number of visits and time before initiation of TPT was minimized. Participants with symptoms were re-assessed 2–3 days later. If CXR was normal and symptoms had resolved, they were offered TPT. If the CXR was abnormal, or symptoms were persistent, one or two sputum samples were sent for

microbiological investigations. If no sputum was obtained, this was considered a negative microbiologic result. Once TBD was excluded, TPT was recommended.

In all three strategies, management decisions, including additional investigations and non-TB related treatment, were made by the sites' clinical staff; research staff acted to remind them of protocol-mandated investigations. If a participant missed a procedure or appointment during the investigation, the study team reminded them, following procedures similar to routine practice. The clinical staff treated participants for TBD and TBI using regimens and follow-up procedures as per National guidelines. If drug-resistant TBD was detected, this was treated per National guidelines.

### TB preventive treatment

In Benin, three months of daily rifampicin and isoniazid (3HR) in fixed-dose combination was used, while in Brazil, 6 months INH (6H), four months of rifampicin (4R) or 12 weekly doses of rifapentine and isoniazid (3HP) were used.

### Outcomes

The primary outcome was the proportion of participants who started TPT in each strategy. Secondary outcomes were the proportions of all participants: completing the strategies for investigation, and detected to have TBD, the proportion of participants starting TPT who completed this, or stopped early because of an adverse event, as well as societal costs per HHC completing each investigation strategy and per HHC completing TPT.

### Data gathering

We gathered baseline microbiologic data and age of index patients and clinical and demographic information of each participating HHC. After randomization, all investigations, treatment given, and visits were recorded. When the investigation of the HHC was completed, or no later than three months post randomization, a summary of investigations and the final diagnosis was completed.

TPT adherence was based on participants' self-report and provider assessments, and treatment outcomes were recorded at the end of therapy. We recorded adverse events that occurred during TPT on a specific case report form.

In February 2022, we amended the original protocol to ascertain incident TB disease that occurred among study participants after completing study procedures and were notified to the National TB program in the two countries. For pragmatic reasons, incidence for all participants in each country was ascertained once, six months after the last participant was enrolled (i.e., in March 2023 in Benin, and May 2023 in Brazil). Follow-up time was calculated as the time from randomization to the date when notifications were reviewed by the National TB program.

### CXR reading

All radiographs were uploaded to an online database, either the original digital files, or for analog films, digitized photos taken of the films. All images were reviewed by the coordinating center, and if technical quality was considered inadequate, these were repeated. Recurrent technical problems at one participating facility resulted in suspension of enrollment until corrected there. Adequate CXR were interpreted by site clinical staff, and classified as: (i) normal, (ii) abnormal not TB, or (iii) abnormal possible TB. Subsequent management of participants was directed by site readers' interpretations. An approximate 10% sample of films were re-read by two expert readers (respirologists with over 30 years of experience with TB care, affiliated with the coordinating center), blind to original site reading, using the same classification system, to verify quality of readings. Agreement between these two expert readers and the site was estimated using non weighted kappa statistics and categorized as follows: Kappa<0 indicates no agreement; Kappa between 0.00 and 0.20 signifies slight agreement; Kappa between 0.21 and 0.40 indicates fair agreement; Kappa between 0.41 and 0.60 reflects moderate agreement; Kappa between 0.61 and 0.80 represents substantial agreement; and Kappa between 0.81 and 1.00 denotes almost perfect agreement [27]. All films from a site were re-read by the expert readers if agreement was poor between site and expert readers; in this circumstance, expert readings were used for the designation of clinically diagnosed TB, used in the analysis.

## Societal (health system plus participants) costs

*Patient costs.* We administered a questionnaire between 1 and 3 months post-randomization, to 120 HHC in Benin and 76 HHC in Brazil, with roughly equal numbers per strategy. This questionnaire was adapted from one used previously to measure patient costs associated with TBD [24], and collected information on out-of-pocket direct costs, as well as time spent (indirect costs) on travel, visits and performing tests. For children, we collected information on the caregivers' time. Indirect costs were valued based on information on the type of work performed by the HHC or caregiver. For Brazil, salaries were gathered from the Brazilian Ministry of Labor and Welfare [28], or the Brazilian Institute of Geography and Statistics [29], or online Brazilian job offer databases. We used the Brazilian minimum wage for unemployed participants, and the average for all salaried participants for self-employed participants. In Benin, we used the GDP [30] for all employed participants, as almost all were self-employed, and the minimum wage for unemployed participants.

**Health system costs.** We recorded all healthcare activities associated with the investigation and management of all HHC's, including protocol-mandated and discretionary activities. Personnel time spent for each type of healthcare activity was estimated from a published study [31], and valued using site-specific salaries provided by facility management. Local unit costs were used to value blood tests, imaging (such as CXR), and microbiological tests (smear, culture, or RMT). The costs of outpatient visits for investigations and TPT were taken from WHO-CHOICE [32].

All costs were collected in the original currency, adjusted for inflation to 2022 values, and then converted to 2022 US Dollars (USD) using direct exchange rates.

## Data analysis and sample size

**Sample size.** We estimated that 50% of HHC would have a positive TST [16] and be eligible for TPT, and, based on a prior trial at the same sites [33,34], in the Standard arm 60% of eligible would start TPT, for an overall TPT initiation rate of 30%. To detect an improvement to an overall TPT initiation rate of 42.5% with 80% power, using a two-sided test, and accounting for clustering of TPT initiation among Household members similar to that seen in an earlier trial [35], we planned to enroll 455 participants into each arm. Allowing for 5% withdrawal, or otherwise not analyzable participants, we inflated this to 478, for a total of 1,434 participants, of whom 1,000 would be enrolled in Benin and 434 in Brazil.

Given this number enrolled to each arm, and using cost estimates from WHO CHOICE database [32] plus data gathered in prior studies [34], we estimated this would provide more than 90% power to detect differences between arms in societal costs within each country.

**Revisions to the sample size estimates.** In January 2021, a study monitor noted that participating HHC in Benin had received reimbursement for travel expenses. Since this reimbursement was not part of standard practice, this was stopped. To assess whether this reimbursement affected study outcomes, in January 2022, we increased the recruitment target from 1,000 to 1,250 in Benin so that approximately half of the participants from Benin received incentives, and half did not. In Brazil, recruitment was significantly impeded by the COVID-19 pandemic. However, more than 99% of participants had analyzable data, not 95% as originally estimated, allowing the recruitment target to be lowered to 370 participants in August 2022.

**Data analysis: Primary.** For the primary analysis, we estimated a risk difference in the proportion of HHC who started TPT within 3 months of randomization among all HHC contacts eligible for TPT between each experimental arm and the standard arm. Treatment initiation was defined as receiving a prescription and dispensed medications for TPT. To test whether there were significant differences in proportion of TPT initiation between each experimental arm and the standard arm, we used logistic regression and Poisson regression, with an identity link, and via generalized estimating equations (GEE) to account for clustering by household. An exchangeable correlation structure and empirical standard errors were used.

**Data analysis: Secondary.**

1. *Completion of protocol-mandated investigations*: We defined each strategy as "completed" if participants underwent all investigations that were specified in the study protocol for that strategy within 3 months after randomization, and "completed

correctly" if participants had no additional tests. We also compared the completion of strategies among participants who did, or did not, receive incentives. Adjusted binomial and Poisson regression models using an identity link and estimated via GEE to account for HHC clustering, with an exchangeable correlation structure and empirical standard errors were used.

2. *Prevalent TB disease*: The prevalence of microbiologically confirmed, or clinically diagnosed TBD detected by the trial procedures within three months of the date of randomization was described; our sample size was not powered to detect a significant difference in prevalent TB disease among strategies. Since all prevalent TBD was diagnosed within 3 months, person time was not considered; instead, simple proportions were compared.

3. *Incident TB Disease* among HHC notified to the NTP after the trial procedures ended, estimated per 100 person years of follow-up, to account for the variable length of follow-up.

4. *Proportion with TPT-related grade 3–4 adverse events* resulting in TPT discontinuation.

5. *Completion of TPT* as defined by the providers and treating teams in each country.

6. *Probable TBI*: In the Standard and RMT arms, we considered HHC with positive TST, or prior TBD, but not prevalent TBD, to have probable TBI. For the No-TST strategy, we first estimated the prevalence of positive TST among HHC allocated to the Standard and RMT strategies stratified by country and age group (5–10, 11–17, 18–24, 25–34 years, and older). This was used to estimate the number of HHC allocated to No-TST Strategy with probable TBI in each age and country strata, and from this the overall number and proportion with probable TBI among all HHC allocated to the No-TST strategy. The proportion was then used in a sensitivity analysis of *costs per probable TBI*.

7. *Societal costs*: Given there was no difference in our primary outcome, we performed a cost-minimization analysis, comparing costs per HHC completing investigation, starting, and completing TPT, in the three strategies. We expressed costs per participant completing investigations, starting and completing TPT, and in a sensitivity analysis, per participant with probable TBI (as defined above) who completed TPT.

We had originally planned to assess sensitivity and specificity of site CXR readings, using the expert readers as a reference standard. However, analysis of the first 100 CXR read by our two expert readers revealed that agreement between them, and between each of them and the site readings was similar enough (95% confidence intervals of kappa values completely overlapping) that we felt we could not use the expert reading as a reference standard and calculate sensitivity and specificity in the traditional manner. We continued to use the expert readers for quality control, as above.

**Ethics.** The study was approved by the Research Ethics Board of the Research Institute of the McGill University Health Centre on August 16, 2019 (Study 2020–5634 Form F1-59915), by the Brazilian National Ethical Board CONEP on May 7, 2020 (#4.014.402), and by the "Comité Local d'Éthique Pour la Recherche Biomédicale (CLERB) de l'Université de Parakou", Benin, on December 30, 2019 (#0293). All HHC provided signed written informed consent. In Brazil, index TB patients also provided signed written informed consent.

A data safety and monitoring board (DSMB) was responsible to review any unusual or unexpected events. The study protocol is registered in clinicaltrials.gov under NCT04528823, and has been published [24].

## Results

### Study population

As summarized in Fig 1, 1589 participants were enrolled, of whom 474 were randomized to the standard arm, 583 to the RMT arm, and 532 to the no-TST arm. In Benin, two participants were enrolled during a site initiation visit between January 29th and February 5th, 2020, then no further enrollment occurred, because of the COVID pandemic, until March 5th, 2021. Enrollment in Benin ended on August 31st, 2022, when enrollment targets were met. In Brazil, enrollment started on

March 27th, 2021, and ended on November 30th, 2022. Registration of the trial, which was to occur in January 2020, at the time of the Benin site initiation visit, was also delayed by COVID disruptions affecting staff at the coordinating center, was submitted on 13 July 2020, and finalized on Aug 27, 2020.

Characteristics of the index cases (Table A in S1 Tables), and of participants were well balanced in the three arms (Table 1). Almost 65% of the participating HHCs were aged 5–24. Of the participants who had a TST, 18% had reactions in the 5–9 mm range and 39% had reactions of ≥10 mm. Participants with a positive TST were somewhat older, and more likely to report drinking alcohol as seen in Table B in S1 Tables. More than 90% of participants in all arms completed protocol mandated investigations correctly; only 6.5% failed to complete study procedures and only 2% had additional tests performed based on clinical staff decisions (Table 2). Of the 745 CXR done, 37 (5.0%) were classified as abnormal-possible TB, with 18 (9.2%), 2 (3.5%), and 17 (3.5%) for the standard, RMT, and no-TST strategies, respectively. As reported elsewhere [36], agreement between the two expert readers on CXR classification was substantial (kappa 0.63 [(95% CI 0.46 to 0.79]), and between site and expert readers was fair [kappa 0.42 (95% CI 0.22 to 0.62), and 0.37 (95%

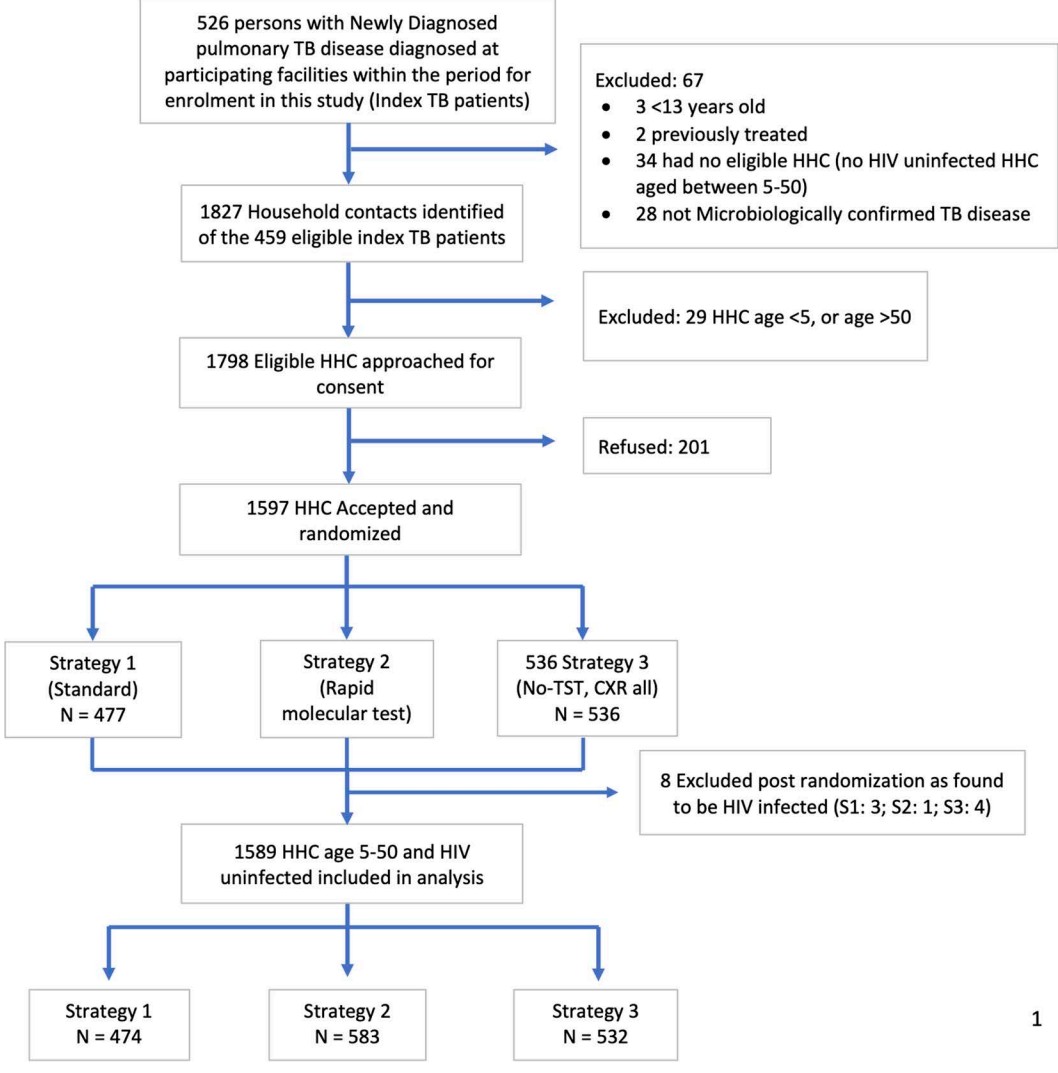

**Fig 1. Consort diagram.**

**Table 1.** Characteristics of household contacts.

| Characteristic (N, and % unless marked otherwise) | Overall | Strategy 1 (Standard) | Strategy 2 (RMT) | Strategy 3 (No-TST) |
|---|---|---|---|---|
| Total number | 1,589 | 474 | 583 | 532 |
| Benin | 1,273 (80.1) | 372 (78.5) | 482 (82.7) | 419 (78.8) |
| Brazil | 316 (19.9) | 102 (21.5) | 101 (17.3) | 113 (21.2) |
| Age group (n, %) | | | | |
| 5–14 | 639 (40.2) | 193 (40.7) | 229 (39.3) | 217 (40.8) |
| 15–24 | 390 (24.5) | 116 (24.5) | 144 (24.7) | 130 (24.4) |
| 25–34 | 263 (16.6) | 68 (14.3) | 112 (19.2) | 83 (15.6) |
| 35–50 | 297 (18.7) | 97 (20.5) | 98 (16.8) | 102 (19.2) |
| Age (median [IQR]) | 18 [11, 29] | 18 [10, 31] | 18 [11, 28] | 17 [10, 29] |
| Male | 748 (47.1) | 232 (48.9) | 278 (47.7) | 238 (44.7) |
| Female | 841 (52.9) | 242 (51.1) | 305 (52.3) | 294 (55.3) |
| Weight, kg (median [IQR]) | 51 [30, 65] | 51 [29, 65] | 51 [32, 66] | 50 [30, 64] |
| Height, m (median [IQR]) | 1.57 [1.37, 1.66] | 1.57 [1.36, 1.66] | 1.57 [1.37, 1.67] | 1.58 [1.40, 1.65] |
| BMI (mean (SD)) | 20.6 (6.1) | 20.7 (6.0) | 20.6 (6.1) | 20.5 (6.2) |
| Major co-morbidities (Diabetes, renal, other) | 8 (0.5) | 3 (0.4) | 1 (0.0) | 4 (0.6) |
| Cigarette smoking: ever* | 72 (4.5) | 18 (3.8) | 33 (5.7) | 21 (3.9) |
| Alcohol—never | 1,069 (67.3) | 302 (63.7) | 420 (72.0) | 347 (65.2) |
| <1 per week | 414 (26.1) | 134 (28.3) | 127 (21.8) | 153 (28.8) |
| ≥1 per week | 106 (6.7) | 38 (8.0) | 36 (6.2) | 32(6.0) |
| Incentives | 665 (41.9) | 197 (41.6) | 238 (40.8) | 230 (43.2) |
| TST performed | 1,019 (64.1) | 453 (95.6) | 563 (96.6) | 3 (0.6) |
| Result: 0–4 mm | 436 (42.8) | 192 (42.4) | 243 (43.2) | 1 (33.3) |
| 5–9 mm | 190 (18.6) | 79 (17.4) | 110 (19.5) | 1 (33.3) |
| ≥10 mm | 393 (38.6) | 182 (40.2) | 210 (37.3) | 1 (33.3) |

*Ever smoker includes current and ex-smokers.

*Abbreviations:* BMI, body mass index; IQR, inter-quartile range; TST, tuberculin skin test.

CI 0.18 to 0.57)]. At one site, the agreement was poor; all films were re-read by the two expert readers. Prevalent TB disease was detected through study procedures in 15 participants, of whom 14 successfully completed treatment.

## Main outcomes: (TST start, investigations completion, TPT completion)

Of those eligible, more than 95% started TPT, of whom 547 (68%) completed TPT. There was no difference in the proportion of TPT initiation between strategies overall (95.2%, 94.2%, and 94.5% for the standard, RMT, and no-TST strategies, respectively), and within 3 months of diagnosis of the index case (92.2%, 92.7%, and 94.3%). Of the 255 who did not complete TPT, only 3 stopped therapy because of an adverse event, and the remainder stopped by patient decision (Table 2). As seen in Table 3, the adjusted risk differences between the two experimental strategies and the standard strategy were not significant for completion of the investigations or for starting TPT. Completion of TPT was 13% lower (95% CI: 3% to 23% lower) in the No-TST strategy compared to the two other strategies. Participants were less likely to start and/or complete if they were aged 25–34, were current cigarette smokers, or drank alcohol (Table C in S1 Tables). The effect of incentives appeared modest—completion of investigations and starting TPT was non-significantly higher with incentives, but completion of TPT lower (Table D in S1 Tables). Incentives did not change the differences between experimental and standard strategies (see adjusted risk differences in Table E in S1 Tables).

**Table 2. Main outcomes: completing investigations, diagnoses, and TPT initiation.**

| Outcomes | Overall | Strategy 1 (Standard) | Strategy 2 (RMT) | Strategy 3 (No-TST) |
|---|---|---|---|---|
| N | 1,589 | 474 | 583 | 532 |
| Completed investigations | 1,485 (93.4) | 439 (92.6) | 534 (95.0) | 492 (92.5) |
| Completed investigations per protocol | 1,448 (91.1) | 422 (89.0) | 541 (92.8) | 485 (91.2) |
| Completed investigations and extra tests | 37 (2.3) | 17 (3.6) | 13 (2.2) | 7 (1.3) |
| Did not complete investigations | 104 (6.5) | 35 (7.4) | 29 (5.0) | 40 (7.5) |
| **Diagnosis* (% of those who completed investigations)** | 1,485 | 439 | 554 | 492 |
| Previous TB disease (treated) | 27 (1.8) | 13 (3.0) | 4 (0.7) | 10 (2.0) |
| Prevalent TB disease** | 15 (1.0) | 3 (0.7) | 6 (1.1) | 6 (1.2) |
| Microbiologically confirmed | 11 (0.7) | 2 (0.5) | 4 (0.7) | 5 (1.0) |
| Clinical diagnosis | 4 (0.3) | 1 (0.2) | 2 (0.4) | 1 (0.2) |
| TB infection (Eligible for TPT) | 848 (57.1) | 166 (37.8) | 206 (37.2) | 476 (96.7) |
| No evidence of TB infection or disease | 595 (40.1) | 257 (58.5) | 338 (61.0) | 0 (0.0) |
| **Treatment (% of those eligible for TPT)** | 848 | 166 | 206 | 476 |
| TPT recommended | 848 (100.0) | 166 (100.0) | 206 (100.0) | 476 (100.0) |
| TPT started (any interval)*** | 802 (94.6) | 158 (95.2) | 194 (94.2) | 450 (94.5) |
| TPT started within 3 months | 793 (93.5) | 153 (92.2) | 191 (92.7) | 449 (94.3) |
| Completed TPT*** (% of started) | 547 (68.2) | 118 (74.7) | 138 (71.1) | 291 (64.7) |
| Did not complete | 255 (31.8) | 40 (25.3) | 56 (28.9) | 159 (35.3) |
| Patient's decision to stop early | 252 (31.2) | 39 (24.7) | 56 (28.9) | 157 (34.9) |
| TPT stopped for adverse events | 3 (0.4) | 1 (0.6) | 0 (0.0) | 2 (0.4) |

*No diagnosis made in those who did not complete investigations.

**All 15 people diagnosed with TB disease started appropriate treatment.

***Number starting and % completing each TPT regimen: 6H: 19 (42%); 3HR: 670 (69%); 4R: 68 (68%); 3HP: 36 (58%).

*Abbreviations*: RMT, rapid molecular test (in this study GeneXpert MTB-Rif); No-TST, no Tuberculin skin test with Chest X-ray for all HHC; TPT, tuberculosis preventive treatment.

The intra-class correlation coefficient, or clustering effect of households, on TPT initiation among study subjects who had at least one other participating family member was 0.49.

## Secondary outcomes: Costs

As seen in Table 4, using the unit costs detailed in Tables F and G in S1 Tables, the per-participant societal costs for completing the evaluation were $74 with the No-TST (universal CXR) strategy compared to $61 with the standard strategy, and $52 with the RMT replacing the CXR. Despite these higher absolute costs, the No-TST strategy had the lowest cost per person initiating TPT ($75 with No-TST, versus $156 with Standard and $134. with RMT) and completing TPT ($136 with No-TST, versus $244 with Standard and $219 with GX). However, after accounting for the likelihood that 315 of the 532 (59%) participants in this strategy would have a negative TST (as shown in Table H in S1 Tables), the cost per person with *probable* TBI starting, and completing TPT was $184 and $331, respectively, with Strategy 3.

## Secondary outcomes: TB disease detection

Finally, as seen in Table 5, prevalent TB disease was detected in 0.9% overall, with the highest proportion of prevalent TB disease detected by the two experimental strategies. The Standard strategy had the highest proportion of participants with TB notified after the study procedures ended, relative to the number detected by study procedures. Of the six participants

**Table 3. Adjusted risk differences in outcomes between experimental (Strategies 1 and 2) and standard (Strategy 1) strategies.** *(Values shown are estimated differences in number of persons with outcome per 100 eligible.).*

| Outcome | Strategy 2 vs. 1 | Strategy 3 vs. 1 | Strategy 3 vs. 2 | Strategy 3 vs. 1 and 2 |
|---|---|---|---|---|
| Completed investigations* | 1.4 (−3.9, 6.8) | 1.1 (−4.7, 6.8) | −0.4 (−5.7, 5) | 0.3 (−4.5, 5.2) |
| Started TPT (% of eligible)** | −5.1 (−9.7, −0.4) | −5.1 (−11.5, 1.4) | 0 (−6.5, 6.5) | −2.2 (−8.4, 4) |
| Started TPT within 3 months (% of eligible)** | −3.6 (−7.1, −0.1) | −1.9 (−6.5, 2.8) | 1.7 (−4, 7.5) | 0.1 (−5, 5.2) |
| Started TPT within 3 months (% of eligible) BENIN only** | −3 (−5.1, −1) | 0 (−1.9, 1.9) | 3 (−1, 6.9) | 1.8 (−1.3, 4.8) |
| Started TPT within 3 months (% of eligible) BRAZIL only** | 1 (−12.4, 14.5) | −6.3 (−23.2, 10.6) | −7.3 (−31.1, 16.5) | −6.7 (−25.8, 12.4) |
| Completed TPT (% of eligible)* | −0.1 (−13.3, 13) | −13.1 (−25.3, −1) | −13 (−24.6, −1.4) | −13.1 (−22.9, −3.2) |
| Completed TPT (% of started)* | 3.5 (−9.4, 16.3) | −11.3 (−23.7, 1) | −14.8 (−26.4, −3.2) | −13.1 (−23.2, −3) |
| Completed TPT (% of started – Benin)*** | 6 (−9.2, 21.1) | −6.8 (−21.4, 7.7) | −12.8 (−25.8, 0.2) | −10.1 (−21.5, 1.4) |
| Completed TPT (% of started – Brazil)*** | 0.8 (−22.8, 24.4) | −24 (−47.4, −0.7) | −24.8 (−49.9, 0.3) | −24.4 (−45.5, −3.3) |

*Notes:* Strategy 1 = standard; Strategy 2 = rapid molecular test; Strategy 3 = no tuberculin skin test and universal Chest X-ray.

An exchangeable correlation structure and empirical standard errors were used for all models.

*Poisson regression model, using an identity link; estimated via generalized estimating equations (GEE) to account for clustering by index TB patient. Adjusted for HHC age, sex, country.

**A binomial regression model, using an identity link, and estimated via generalized estimating equations (GEE) to account for clustering by site.

***Binomial regression model, using an identity link; estimated via generalized estimating equations (GEE) to account for clustering by index TB patient, adjusted for HHC age and sex.

*Abbreviations:* HHC, household contact; RMT, rapid molecular test (in this study GeneXpert MTB-Rif); No-TST, no Tuberculin skin test, with CXR in all HHC; TPT, tuberculosis preventive treatment.

who were notified to have microbiologically confirmed TB disease after study procedures, five were enrolled in Benin, and were considered to have a negative TST, of whom two had a TST of 5–9 mm; none had a second TST 8 weeks after exposure to detect late conversion. Of the participants randomized to the RMT strategy who could not produce a sputum sample, none were diagnosed at the time, or subsequently, to have TB disease through other diagnostic methods.

## Details of persons with Incident TB disease

### Strategy 1 (Standard).

Participant 1: 38 years old, in Benin, TST 6 mm (cut off in Benin >10 mm), CXR not recommended, TPT not recommended. Diagnosed, 5 months after randomization, micro–confirmed (GeneXpert)

Participant 2: 10 years old, in Benin, TST 0 mm, CXR not recommended, TPT not recommended, 17 months after randomization, micro–confirmed (GeneXpert)

Participant 3: 12 years old, in Benin, TST 0 mm, CXR not recommended, TPT not recommended, 17 months after randomization, micro-confirmed (GeneXpert)

Participant 4: 18 years old, in Brazil, TST 5 mm (cut off in Brazil > 5 mm), Normal CXR, refused TPT, diagnosed 5 months after randomization, micro-confirmed (GeneXpert and Culture positive), died due to TB.

### Strategy 2 (RMT).

Participant 5: 50 years old, in Benin, TST 0 mm, GeneXpert not recommended, TPT not recommended, 3.5 months after randomization, clinically confirmed (CXR)

Participant 6: 39 years old, in Benin, TST 6 mm, GeneXpert not recommended, TPT not recommended, 16 months after randomization, clinically confirmed (CXR)

**Table 4. Health system and participant costs—overall and per outcome. (All costs in 2022 US dollars).**

| Characteristic/cost | Strategy 1 (Standard) | Strategy 2 (RMT) | Strategy 3 (No-TST) |
|---|---|---|---|
| Total number of participating HHC | 474 | 583 | 532 |
| Total societal costs | 28,768.94 | 30,284.79 | 39,516.01 |
| Total health system costs | 21,537.18 | 23,472.28 | 33,354.25 |
| Total participant costs: | 7,231.76 | 6,812.51 | 6,161.75 |
| Direct | 3,331.71 | 3,060.89 | 2,288.65 |
| Indirect (time) | 3,900.05 | 3,751.62 | 3,873.10 |
| ***Total Costs per HHC*** | | | |
| Total societal costs per HHC | $60.69 (56.41, 64.98) | $51.95 (48.88, 55.01) | $74.28 (72.03, 76.53) |
| Total health system cost per HHC | 45.44 | 40.26 | 62.7 |
| Total participant costs per HHC | 15.26 | 11.69 | 11.58 |
| ***TB investigation costs, excluding TPT treatment*** | | | |
| Societal costs | 20,662.14 | 21,379.28 | 23,620.78 |
| Health system costs | 15,350.05 | 16,100.00 | 20,361.20 |
| Participant costs: | 5,312.09 | 5,279.28 | 3,259.58 |
| Direct | 2,642.49 | 2,555.11 | 1,274.57 |
| Indirect (time) | 2,669.60 | 2,724.17 | 1,985.01 |
| ***Cost per HHC with prevalent TB disease detected**** | | | |
| Number with prevalent TB disease detected in trial | 3 | 6 | 6 |
| **Societal** cost per TB disease detected* | $6,887(0, 14,676) | $3,563 (719, 6,407) | $3,937 (800, 7,073) |
| Health system cost per TB disease detected* | 5,116.68 | 2,683.33 | 3,393.53 |
| Participant cost per TB disease detected* | 1,770.70 | 879.88 | 543.26 |
| ***Cost per HHC starting TPT**** | | | |
| Number of HHC starting TPT (any time) | 158 | 194 | 450 |
| Total **societal** cost per HHC starting TPT)** | $156 (133, 178) | $133.95 (117, 151) | $75.42 (72, 79) |
| Total health system cost per HHC starting TPT** | $122.15 | $106.73 | $68.17 |
| Total participant cost per HHC Starting TPT** | $33.62 | $27.21 | $7.24 |
| ***Cost per HHC completing TPT***** | | | |
| Number of HHC completing TPT | 118 | 138 | 291 |
| Total societal cost per HHC completing TPT*** | $243.80 (202, 286) | $219.45 (185, 254) | $135.79 (125, 147) |
| Total health system cost per HHC completing TPT*** | $182.52 | $170.09 | $114.62 |
| Total participant cost per HHC completing TPT*** | $61.29 | $49.37 | $21.17 |
| ***Cost per HHC with documented or estimated TBI#*** | | | |
| Number of HHC with **assumed TBI** starting TPT | 158 | 194 | 184.5 |
| Total **societal** cost per HHC starting TPT)** | $155.77 | $133.95 | $183.95 |
| Number of HHC completing TPT | 118 | 138 | 119.3 |
| Total **societal** cost per HHC completing TPT)*** | $243.80 | $219.45 | $331.22 |

*Calculation of costs per TB disease detected, was restricted to costs up to detecting TB, meaning all investigation costs, but no TPT costs. Costs for treating the TB disease were not included—i.e., strictly costs to detect TB disease.

**Costs per HHC starting TPT includes all investigation, plus TPT-related medications.

***Costs per HHC completing TPT includes all investigation, plus all TPT-related costs (medications and follow-up).

#TB infection assumed in HHC with positive TST. Prevalence of positive TST in Strategy 3 estimated from age and country stratified prevalence of positive TST among HHC who had TST as part of investigations in Strategy 1 and 2 (shown in Table H in S1 Tables).

*Abbreviations:* HHC, household contact; RMT, rapid molecular test (in this study GeneXpert MTB-Rif); No-TST, no Tuberculin skin test with Chest X-ray for all HHC; TPT, tuberculosis preventive treatment.

**Table 5. TB Disease, detected through study procedures, and notified subsequently.**

| | Overall | Strategy 1 (Standard) | Strategy 2 (RMT) | Strategy 3 (No-TST) |
|---|---|---|---|---|
| Number of HHC | 1,589 | 474 | 583 | 532 |
| **Prevalent TB disease** (*detected through study procedures*) | | | | |
| Total prevalent TBD (*N*, % of all HHC) | 15 (0.9) | 3 (0.6) | 6 (1.0) | 6 (1.1) |
| Confirmed | 11 (0.7) | 2 (0.4) | 4 (0.7) | 5 (0.9) |
| Clinical diagnosis | 4 (0.3) | 1 (0.2) | 2 (0.3) | 1 (0.2) |
| **Incident TB Disease** (*Notified to TB program after study procedures completed*) | | | | |
| Total incident TBD (*N*, % of all HHC) | 6 (0.4) | 4 (0.8) | 2 (0.3) | 0 |
| Incidence of TBD per 100 PY (95% CI) | 0.38 (0.14, 0.82) | 0.86 (0.23, 2.1) | 0.35 (0.04, 1.25) | 0 (0.00, 0.68) |
| **Total TB disease**: *Prevalent TBD detected through study procedures and incident TBD notified later* | | | | |
| Total prevalent and incident TBD (*N*, % of all HHC randomized) | 21 (1.3) | 7 (1.5) | 8 (1.3) | 6 (1.1) |
| Percent of total with TBD detected by study procedures | 71% | 43% | 75% | 100% |

*Abbreviations:* HHC, household contact; RMT, rapid molecular test (in this study GeneXpert MTB-Rif); No-TST, no Tuberculin skin test, with chest X-ray for all HHC; TPT, tuberculosis preventive treatment.

## Discussion

In this trial of 1,589 HHC aged 5–50 in Benin and Brazil, 93% completed all investigations, and of those eligible, 95% initiated TPT, although only 68% of those completed TPT. There was no difference among strategies in the proportion completing investigations and starting TPT, although the No-TST strategy was associated with lower TPT completion. Total societal costs per HHC completing evaluation were lowest with the RMT strategy, and highest with the No-TST (universal CXR) strategy. On the other hand, costs were lower per TPT completed with the No-TST strategy. However, when we accounted for the likelihood of TB infection among participants in this strategy, the societal cost per person with 'probable TB infection' who started or completed TPT was substantially higher than the other two arms.

Strengths of the study include the pragmatic design that tested simple interventions available in the two countries, and randomization to minimize potential selection bias for testing and management strategies. As well, several relevant outcomes were measured, notably detection of TB disease, initiation and completion of TPT, and costs from health system and participant perspectives.

Nevertheless, there were a number of important potential limitations. One was the likely effect of the study itself on provider and HHC behavior, given that 93% of participants completed study procedures. In addition, almost half the participants in Benin received some form of compensation to defray costs associated with visits, although we did not find a significant difference in participant outcomes associated with receipt of these reimbursements. The small number of HHC detected with prevalent and incident TB disease limited power and precluded statistical testing of differences between arms. As well this study was conducted during the COVID pandemic. This delayed and impeded enrollment, particularly in Brazil, preventing attainment of final sample size required there. HIV status was self-reported for HHC's, which may have lead to some inaccuracy, although HIV testing was routine for HHCs if they were sexual partners of an HIV-positive index TB patient. In Benin, all HHCs were given TPT medications in the clinics where they were seen. In Brazil, this procedure was followed in some clinics, but in others HHCs were given a prescription which they filled at a pharmacy. These different procedures may have affected actual TPT starting, but these different approaches were balanced across arms, and we measured completion, to account for any differences in HHCs actually receiving and starting TPT medications.

Perhaps the most interesting finding of this study was the high rate of completion of investigations, and initiation of TPT among HHC judged eligible, in all three arms. This finding is particularly relevant for the two arms incorporating TST, given

the current debate over the need for this, due to the perceived added complexity. Although this high rate of completion of investigations may have reflected the "study effect' as discussed above, this may also have reflected efforts at all participating facilities to provide person-centered care as suggested by WHO [25]. We also limited the investigation of symptoms to persons who reported symptoms on the first visit which were **persistent** on the day of TST reading. Global access to tuberculin testing material has been difficult with limited supplies in many countries. The new MTB-specific tuberculins offer substantial promise given their improved specificity plus the current prices of $1.50 per dose if purchased through the Global Drug Facility.

Another interesting finding was the feasibility and acceptability of the RMT strategy, given that almost 95% of HHC completed all investigations in this arm. The lower total health system costs per participating HHC would make this strategy appealing to public health officials and policymakers, while the lower participants' costs compared to the standard evaluation would be attractive for patients. The diagnostic yield appeared comparable to the other strategies. A concern with this strategy was that 44 participants could not produce spontaneous sputum samples, leading to potential underdiagnosis of TB disease in this arm. However, TB disease was not detected by other methods, nor was incident TB declared after, although inferences are limited, given the small number of participants.

The strategy without TST, despite being simpler, was not associated with a significant improvement in the cascade of care, and this strategy had higher health system costs per HHC investigated. On the other hand, an important advantage for participants was that their costs were lowest with this strategy, reflecting the reduced number of visits required. The finding that no participants in this arm were detected with incident TB later, suggests that screening all HHCs for TB disease may help detect more HHCs with minimal prevalent disease that will otherwise be missed. However, this finding must be interpreted very cautiously given the small numbers and post hoc assessment. A much higher proportion of enrolled HHCs were started on TPT, making the societal costs per HHC starting or completing TPT lowest with this strategy. However, if the cost of TPT medications and follow-up is higher in other settings, this may not be the least costly approach. In addition, given that the prevalence of positive TB infection tests was only 41% in the other two arms it is likely that the majority of the participants receiving TPT with this strategy would not have derived benefit, based on evidence from prior systematic reviews of placebo controlled trials of TPT in PLHIV [12], other trials [13–15] and IPD meta-analyses [10,11]. Gupta and colleagues found a prevalence of 72% positive TBI tests among HHCs of MDR index patients and concluded that TBI testing could be skipped for this group [37]. However, in our study, the prevalence of probable TBI was much lower (41%) in HHC of drug-susceptible index patients. This difference may reflect more prolonged exposure from MDR index patients and emphasizes the importance of obtaining relevant epidemiologic information to inform discussions about the value of TBI testing of HHC. When we accounted for the lower probability of TB infection, costs per person starting and completing TPT were highest with this strategy. A further consideration is the significantly lower TPT completion rate in this arm, which may have reflected lower motivation.

We conclude that in Benin and Brazil, more than 90% of eligible HHC completed investigations, and initiated TPT with WHO-recommended strategy, or with two experimental strategies for management of HHC. The experimental strategy of replacing chest X-ray with RMTs appears feasible, with acceptable yield, and lower costs.

## Supporting information

**S1 Fig. Schematic of the three strategies.**
(DOCX)

**S1 Tables. Supplemental Tables.** Table A: Summary of Index TB patients. Table B: Characteristics of HHC associated with Positive TST (among tested, in Strategies 1 and 2). Table C: Descriptive analysis of characteristics of participants who started and completed TPT. Table D: Unadjusted estimates of effect of incentives on outcomes. Table E: Adjusted risk differences in outcomes between experimental Strategies 2 and 3, and standard Strategy 1, stratified by receipt of

incentives. Table F: Detailed health system costs in Benin and Brazil, in 2022 USD. Table G: Costs of drugs used for TPT, from 2022 GDF catalogue. Table H: Prevalence of positive TST stratified by country and age group, used to estimated number with TB infection (TBI) among HHC randomized to Strategy 3 (No-TST).
(DOCX)

**S1 CONSORT Checklist.**
(DOCX)

**S1 Protocol.**
(DOCX)

**S1 Statistical Analysis Plan.**
(DOCX)

## Acknowledgments

We thank the *Laboratoire de Télématique Biomédicale du Réseau en Santé Respiratoire du Québec* (https://gxt-ltb.cred.ca/) that developed the web-based programme and housed data on registration, eligibility, randomization, and all data of participants.

## Author contributions

**Conceptualization:** Dick Menzies.

**Funding acquisition:** Dick Menzies.

**Methodology:** Menonli Adjobimey, Anete Trajman, Mayara Lisboa Bastos, Diana Gibson, Frimege Djohoun, Olivia Oxlade, Federica Fregonese, Dissou Affolabi, Valentin Kouchade, Elisa Aguiar, Renata Spener-Gomes, Marcelo Cordeiro-Santos, Renato T Stein, Marcelo Scotta, Andrea Benedetti, Dick Menzies.

**Supervision:** Dick Menzies.

**Writing – original draft:** Menonli Adjobimey, Dick Menzies.

**Writing – review & editing:** Menonli Adjobimey, Anete Trajman, Mayara Lisboa Bastos, Chantal Valiquette, Diana Gibson, Frimege Djohoun, Olivia Oxlade, Federica Fregonese, Dissou Affolabi, Valentin Kouchade, Elisa Aguiar, Renata Spener-Gomes, Marcelo Cordeiro-Santos, Renato T Stein, Marcelo Scotta, Andrea Benedetti, Dick Menzies.

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
