## [Editor Report · Decision Letter 0]

Dear Dr Menzies,

Thank you for submitting your manuscript entitled "Rapid molecular testing or chest-X-ray or tuberculin skin testing for household contact assessment: Results of a cluster-randomized trial." for consideration by PLOS Medicine.

Thank you also for your patience whilst we asked about the reasons behind the late registration of the study. Your manuscript has now been evaluated by the PLOS Medicine editorial staff as well as by an academic editor with relevant expertise and I am writing to let you know that we would like to send your submission out for external peer review.

Please re-submit your manuscript within two working days, i.e. by Oct 31 2024 11:59PM. Please do let us know if you need more time.

Kind regards,

Syba

Syba Sunny, MBBS, MRes, FRCPath

Associate Editor

PLOS Medicine

---

## [Decision Letter · Decision Letter 1]

Dear Dr Menzies,

Many thanks for submitting your manuscript "Rapid molecular testing or chest-X-ray or tuberculin skin testing for household contact assessment: Results of a cluster-randomized trial" (PMEDICINE-D-24-03440R1). The paper has been reviewed by two subject experts and a statistician; their comments are included below and can also be accessed here: [LINK]

As you will see, the reviewers were supportive of the study but raised a number of questions and clarifications about the methodology. After discussing the paper with the editorial team and an academic editor with relevant expertise, I'm pleased to invite you to revise the paper in response to the reviewers' comments. We plan to send the revised paper to some or all of the original reviewers, and we cannot provide any guarantees at this stage regarding publication.

When you upload your revision, please include a point-by-point response that addresses all of the reviewer and editorial points, indicating the changes made in the manuscript and either an excerpt of the revised text or the location (eg: page and line number) where each change can be found. Please also be sure to check the general editorial comments at the end of this letter and include these in your point-by-point response to the editors. When you resubmit your paper, please include a clean version of the paper as the main article file and a version with changes tracked as a marked-up manuscript. It may also be helpful to check the guidelines for revised papers at http://journals.plos.org/plosmedicine/s/revising-your-manuscript for any that apply to your paper.

In view of the upcoming holiday, we ask that you submit your revision by Monday, January 6th. However, if this deadline is not feasible, please contact me by email, and we can discuss a suitable alternative. Please also feel free to contact me with any questions (hvanepps@plos.org).

Kind regards,

Heather

Heather Van Epps, PhD

Executive Editor

[on behalf of]

Syba Sunny, MBBS, MRes, FRCPath

Senior Editor

PLOS Medicine

ssunny@plos.org

Comments from the editors:

1. Please ensure that your abstract is formatted according to CONSORT for Abstracts. Details are included in the general editorial points outlined below.

2. Please add the dates of recruitment (month, day, year) to the abstract.

3. Our general rule for secondary outcomes in the abstract is that they should all be included or none should be included (ie, report only the primary outcome, per CONSORT). If you wish to include secondary outcomes in the abstract, you should report the results for each of them. Given the number of secondary outcomes included in the trial, our preference would be for your abstract to report only the primary outcome.

4. Your current Data Availability statement requires additional detail and explanation. The PLOS Data Availability Policy is here: http://journals.plos.org/plosmedicine/s/data-availability). In brief, we require that all data underlying the study's findings be provided in a repository or as Supporting Information. For data residing with a third party, authors are required to provide instructions with contact information (web or email address) for obtaining the data. Please note that a study author cannot be the contact person for the data. Given that these are not third party data, we would require you to provide the minimal data set within the paper; ie, the data required to replicate the findings of a study, along with related metadata and methods. This includes the dataset used to reach the conclusions in the manuscript, any additional data needed to replicate the study findings, and related metadata and methods.

5. Tables: Please use standard formatting for footnotes, removing labels (eg, abbreviations, notes).

Comments from the reviewers:

Reviewer #1: Statistical review

This trial reports a cluster randomised trial comparing different strategies for testing contacts of TB diagnosed individuals. Generally the trial appears to have been conducted well and provides interesting data. I have some comments on the statistical methods and reporting, provided below:

1. Abstract: I would recommend that the primary outcome proportion is provided for each group in addition to what is currently reported.

2. Interventions: As a non-expert, I did not quite follow how the experimental arms would be expected to improve the TPT initiation rate (which would presumably be the case with a superiority trial). Could this be spelt out a bit more?

3. Methods: I would recommend an 'Outcomes' subsection is created where all the primary and secondary outcomes are listed. I note later that these are laid out in the analysis section, so perhaps an alternative is to add 'Outcomes' to that subsection title.

4. Sample size calculation: I assume the power calculation used a two-sided 5% type I error rate: could this be added? Also, was clustering allowed for? Regardless, it would be useful to estimate the clustering effect and provide this for the benefit of future sample size calculations.

5. Data analysis - secondary: it wasn't clear to me what analysis method was used for Incident TB Disease among HHC to allow for the variable follow-up.

6. Results - the sample size calculation is written as if the main outcome is the proportion of those randomised who initiated TPT, but the results are written as the proportion of those who were eligible. I see the registration page uses the latter, but isn't the former also a valid randomised comparison to do?

7. Table 4 - is there a way of getting uncertainty estimates for the differences in cost-effectiveness between arms? It would be interesting to know whether observed differences are beyond that consistent with variability. If this is possible, I would add this to the abstract too.

James Wason

Reviewer #2:

In this study, Adjobimey and colleagues conducted an RCT in Benin and Brazil to investigate three different strategies for managing household contacts of people with TB. Approaches included combinations of symptom screening, TST, chest x-ray, and rapid molecular testing. The primary outcome was initiation of TPT, which was found to be similar across the three arms. The inclusion of a costing analysis as part of the initial manuscript disseminating trial results is quite novel in TB, and is a relevant inclusion for a paper focused on, essentially, modes of preventing TB. This seems to be a well-conducted study with many findings of interest to the field. Some specific comments for consideration are below.

Major comments:

1. Methods, CXR reading: was there a pre-specified plan stating definitions of good or poor agreement between readers? How 'poor' would agreement need to be before an expert reader had to review all CXRs? Could a short explanation addressing this be included in this paragraph? - updated - From reading the results, seems a kappa-statistic was calculated (regular or weighted or other method, unsure) - I think this could be more clearly indicated in the methods section. Was agreement mainly influenced by a high proportion of 'normal' CXRs? Including positive percent agreement and negative percent agreement in results could clearly indicate where reviewers agreed. Is the number of CXRs in each category (abnormal-not TB/abnormal-possible TB/normal) ever stated in the paper? If not, can this be included in the paper/supplement somewhere?

2. Methods, Data analysis - Secondary: 3. Incident TB Disease: I am happy to be convinced that the described method is the best approach to estimating TB disease incidence rate. But I also wanted to check whether the author team had considered that the TBI risk may not be uniform across time, and therefore summarising measures per 100 person-years may not be the ideal approach? Would it perhaps be informative to also perform a survival analysis, stratify based on experimental strategy, and run a test to check for statistically-significant differences between curves/arms? This ultimately may not tell us much due to relatively low frequency of events, but assumedly since authors have follow-up times available it could be incorporated into analysis.

Minor comments:

3. Title: the title actually doesn't really mention TB/TBI - suggest modifying so non-TB people have a clearer idea of what the study is focussed on.

4. Abstract, Background: it is stated that WHO recommends evaluation of all HHCs, but should clearly specify what the evaluation is actually for, i.e., the use-case. As stated, this paragraph is a little bit vague.

5. Abstract, Methods: I think it needs to be mentioned somewhere here that HHCs' index case had drug-susceptible TB (i.e., drug-resistant TB excluded).

6. Introduction, first paragraph: as 2024 WHO Global TB Report has now been published, would recommend updating figures to be consistent with new report.

7. Introduction, third paragraph: author team writes that 'Provision of TPT to persons with undetected TB disease could result in acquired drug resistance' - but link to preceding sentences isn't fully clear. I think this is supposed to indicate that TPT is, unlike in the trial context, frequently given to people who have not had TBD excluded. Would strongly suggest making this more explicit in-text (with a citation), to strengthen the argument and relevance of the potential (and possibly common) harm that may arise from TPT.

8. Introduction, fourth paragraph: suggest spelling out the interventions/experimental evaluations in each arm, as this is a statement of the study's main objective.

9. Methods, Study sites: further information about selected clinics would be helpful for understanding the 'kind' of index cases recruited, and in-turn, the risk their TB poses to their household contacts (which may be relevant when considering generalisability of findings). Could some detail about clinics be included in the supplement? Were these outpatient clinics at general hospitals? District level health centres? Outpatient clinics of respiratory hospitals? Something else? As well, does 'diagnostic facilities' refer to the diagnosis of TB disease or TB infection?

10. Methods, Experimental 1 - rapid molecular tests: confirming whether the Xpert test used was Xpert MTB/RIF or Xpert MTB/RIF Ultra?

11. Methods, Data analysis - primary: Can you please state the primary outcome measure here? It's stated in abstract but not here really. As well, suggest rewording this paragraph to be a bit more specific and indicate that testing was for significant differences in *outcome* between arms, rather than 'significant differences between experimental arms'. When I first read this section, I did not understand what the outcome variable was - only when I read the results did I realise a risk difference was being calculated.

12. Results, above Table 2: exactly what was the proportion of people initiating TPT in each arm? Since this is most related to the primary outcome, suggest describing these figures in-text somewhere.

13. Table 2: I think this table contains all the relevant information, but it needs to be tuned. What exactly does the row header 'Completed' mean? On first glance, I thought it meant TPT was completed, but I think it must be referring to study procedures. Is there a way to demarcate the values used to calculate the primary analysis (i.e., 'proportion of HHC who started TPT within 3 months of randomization among all HHC contacts eligible for TPT'.)

14. Discussion: very thorough and contextualises the results for both healthcare systems and patients. The explanations about how the trial staff made the intervention 'patient-centred' were particularly welcome.

Reviewer #3:

The manuscript presents a well-conducted and relevant study addressing the complexities of TB preventive treatment (TPT) implementation across varying settings. The abstract requires no changes, but the introduction needs updated references and expanded reasoning on disparities in TPT target achievement among populations, particularly PLHIV versus other groups. Some methodological clarifications are needed, including the handling of drug resistance testing, HIV status confirmation, and intervention-specific details on chest X-ray (CXR) and microbiological methods. Results would benefit from inclusion of HIV-positive participant proportions and Number-Needed-to-Test (NNT) data for strategic comparisons. The discussion is strong but could delve deeper into the role of initial screening, costs of household contact investigation, and the implications of simplified strategies on health systems. Additionally, comparing findings with Gupta et al. (2020) could enhance insights on feasibility and scalability. Addressing challenges with TB infection testing and reflecting on WHO-approved TB-specific skin tests would further strengthen the manuscript's practical relevance.

Specific comments:

Abstract: No changes suggested

Introduction:

1. Update reference to the Global TB report as the 2024 version is now available.

2. It would be helpful to expand the reasoning for WHY/HOW was the TPT targets achieved among PLHIV, but not for other groups?

3. In paragraph 2, there is mention of WHO not requiring the use of TBI testing or CXR, but TPT started based on risk group classification - this sentence is misleading, as the guidance still requires ruling out TBD or at least not having clinical suspicion for TB.

4. Use of PLHIV and PHIV - be consistent with abbreviations.

Methods:

1. Within the definition of index patients, the abbreviation for GeneXpert is used i.e. GX; please spell this out first before using the abbreviation. Also, clarify if this was GeneXpert MTB/Rif or Ultra.

2. Among index patients, how was drug resistance ruled out among those diagnosed on smear or culture - was additional reflex testing done to rule out such patients as having additional drug resistance; please clarify.

3. Among those HHC with a known HIV status (they were not tested by the study), how was the status confirmed; this is limitation since some (with an unknown status) would have been tested while the 'other group' would have been largely self-reported.

4. As described within the 'Standard' intervention, was CXR read once (and a decision made) or were multiple reads conducted; it is assumed that this was done by human-readers and not aided by computer-aided detection software, however this should be explicitly stated. In addition, please clarify what microbiological tests were done.

5. Was the same TPT used in all interventions? Currently this is described within the 'Experiment 2' paragraph, which suggests it only applied to this intervention; might help to have this separated if applied to all interventions.

6. In the data analysis, treatment initiation was defined as receiving a prescription, or dispensed medications for TPT - these are two quite different prospects in relation to 'initiating treatment' i.e., pills in hand are close to initiation treatment versus having to still collect medication; is there a reason why this was done? Would be helpful to understand the rationale for allowing both of these scenarios.

Results:

1. The proportion of participants who were HIV positive is not shown in any table (including the supplemental ones); given the relevance of this population for TPT, this should be highlighted to understand its context in these geographies.

2. Given the variations in the strategies i.e., the entry point for TBD testing, it would be helpful to also show the Number-needed-to test (NNT) for each strategy. This would complement the costing perspective as well.

Discussion:

1. Interesting study and clearly presented.

2. The topic of TPT uptake is an important one and complex given the variation of approaches deployed across geographies and interpretation of WHO guidance. What is becoming clearer through studies like this one, is that simpler strategies (that rule of TBD) might be better.

3. In your study, your results showed no incident TB cases were reported in the No-TST arm compared to the other arms - does this make the point that better initial screening (with CXR) helps to ensure that no cases of TB were missed during contact investigation? Would suggest exploring this point further in the discussion.

4. Household contact investigation is a costly intervention, including the need for multiple visits by HHCs (to healthcare facilities) or healthcare staff to households. This is further compounded by TST testing (where multiple visits to read induration are required) - is there data available to show the number of household visits conducted to complete the TBI investigation, and if this influenced the health system or participant costs?

5. Gupta et al (Clinical Infectious Diseases; 2020) reported an interesting finding in a feasibility study conducted to inform the development of a TPT interventional trial where they assessed eligibility of HHCs of DR-TB index patients for TPT based on their classification per WHO guidance (TBI status, <5 years, or having HIV infection). They reported among 1007 HHCs, 775 (77%) were considered high-risk per these mutually exclusive groups. This highlighted the possibility that given the high proportion eligible, HHCs should be started on TPT by focusing on ruling on TB disease rather than performing other investigations, like TB infection testing. In essence, it made the case for simplifying the intervention in TB households. Importantly, these were DR-TB households, and the TB infection prevalence (72%) was quite high, and likely to be different in DS-TB households. It would be helpful to reflect on this study and its relevance given what you have found with your findings - does this support a similar approach? Does your HHC data look similar?

6. There are known challenges associated with implementing TST testing - was this observed in the study? Given the recent WHO-approval of TB specific skin tests, and their low costs (~$2), could this change the feasibility and costs associated with TB infection testing as implemented in your study?

7. PLHIV was noted in the introduction as the one high-risks groups (as per WHO) where UNHLM TPT targets were achieved - there is no specific discussion or data presented on this and its relevance in your study; what does this suggest for low HIV-prevalence settings?

---

* Please upload any figures associated with your paper as individual TIF or EPS files with 300dpi resolution at resubmission; please read our figure guidelines for more information on our requirements: http://journals.plos.org/plosmedicine/s/figures. While revising your submission, please upload your figure files to the PACE digital diagnostic tool, https://pacev2.apexcovantage.com/. PACE helps ensure that figures meet PLOS requirements. To use PACE, you must first register as a user. Then, login and navigate to the UPLOAD tab, where you will find detailed instructions on how to use the tool. If you encounter any issues or have any questions when using PACE, please email us at PLOSMedicine@plos.org.

* As above, please ensure that the paper adheres to the PLOS Data Availability Policy (see http://journals.plos.org/plosmedicine/s/data-availability), which requires that all data underlying the study's findings be provided in a repository or as Supporting Information. For data residing with a third party, authors are required to provide instructions with contact information (web or email address) for obtaining the data. Please note that a study author cannot be the contact person for the data. PLOS journals do not allow statements supported by "data not shown" or "unpublished results." For such statements, authors must provide supporting data or cite public sources that include it.

* Please ensure that the study is reported according to the CONSORT for cluster randomized trials guideline and include the completed CONSORT checklist as Supporting Information (https://www.equator-network.org/reporting-guidelines/consort-cluster/). When completing the checklist, please use section and paragraph numbers, rather than page numbers. Please add the following statement, or similar, to the Methods: "This study is reported as per the CONSORT for cluster randomized trials guideline (S1 Checklist)."

FIGURES AND TABLES

SUPPLEMENTARY MATERIAL

REFERENCES

Randomized controlled trials

* Please structure the Methods section using the following sub-headings: Study design and participants, Randomization and masking, Procedures, Outcomes, Statistical analysis.

* Please ensure that the outcomes in the submitted manuscript match those outlined the protocol [and/or trial registry]. Please clarify and explain all discrepancies between the paper and protocol. If the outcomes were not prespecified in the protocol, please define them in the Methods (Outcomes section) as post hoc and explain why they were added. Post-hoc comparisons should be presented as hypothesis generating rather than conclusive.

* Please ensure that all prespecified outcomes (primary, secondary, and exploratory) are listed in the Methods/Outcomes section and indicate whether there are outcomes that are not presented in the current report.

* Please specify the dates (Month, Day, Year) during which study enrollment and follow up occurred.

* Please include absolute numbers wherever you report percentages; eg, n/N (%)

* Please complete the CONSORT checklist (https://www.equator-network.org/reporting-guidelines/consort/) and ensure that all components of CONSORT are present in the manuscript, including how randomization was performed, allocation concealment, blinding of intervention, definition of lost to follow-up, power statement. When completing the checklist, please use section and paragraph numbers, rather than page numbers.

* Please report your abstract according to CONSORT for abstracts, following the PLOS Medicine abstract structure (Background, Methods and Findings, Conclusions) https://www.equator-network.org/reporting-guidelines/consort-abstracts/

* If your trial had to undergo important modifications in response to extenuating circumstances, please complete the CONSERVE-CONSORT checklist and provide in your Supporting Information; (https://www.equator-network.org/reporting-guidelines/guidelines-for-reporting-trial-protocols-and-completed-trials-modified-due-to-the-covid-19-pandemic-and-other-extenuating-circumstances-the-conserve-2021-statement/). When completing the checklist, please use section and paragraph numbers, rather than page numbers.

* In keeping with our commitment to Open Science, please include the study protocol document and analysis plan (including any amendments) as Supporting Information to be published with the manuscript if accepted.

* Please note that PLOS Medicine requires prospective, public registration of a data sharing plan (as part of mandatory clinical trials registration) for all clinical trials that began enrollment on or after January 1, 2019, in accordance with ICMJE requirements.

---

## [Decision Letter · Decision Letter 2]

Dear Dr. Menzies,

Thank you very much for re-submitting your manuscript "Rapid molecular testing or chest-X-ray or tuberculin skin testing for household contact assessment of tuberculosis infection: Results of a cluster-randomized trial" (PMEDICINE-D-24-03440R2) for review by PLOS Medicine.

I have discussed the paper with my colleagues and the academic editor and it was also seen again by three reviewers. I am pleased to say that provided the remaining editorial and production issues are dealt with we are planning to accept the paper for publication in the journal.

[LINK]

Please note that the data availability issue needs to be sorted prior to acceptance.

We look forward to receiving the revised manuscript by Mar 27 2025 11:59PM.   

Sincerely,

Alison Farrell, PhD

Senior Editor 

PLOS Medicine

plosmedicine.org

Requests from Editors:

Title:

Please revise according to PLOS Medicine format. “Results” should not be in the title.

E,g, Comparison of rapid molecular testing, chest-X-ray and tuberculin skin testing for household contact assessment of tuberculosis infection: a cluster-randomized trial.

Abstract:

The Abstract consists of the following subheadings: Background, Methods and findings, Conclusions

Please use the active voice throughout, including in the Abstract. E.g. We conducted a superiority, open-label, cluster-randomized trial….

Please clarify what is meant by “completion of investigations” as a secondary outcome. Please revise. Please specify the secondary outcomes as per the protocol and report on the findings for all.

Please clarify “Of 848 eligible” (add HHC).

Please remove the term ‘high-quality’ from the Abstract. Please include a sentence on study limitations. Please clarify whether, in the penultimate sentence of the abstract, you mean by “TBI testing” tuberculin skin testing/standard testing?

Please check compound adjectives, e.g. resource-limited?protocol-mandated? throughout the manuscript.

Introduction:

TPT is defined as preventive TB treatment. Should this be TB preventive treatment? Also, TPT should be explained (i.e. the general reader is unaware that it is a monotherapy until the last paragraph of the Intro).

Please break penultimate sentence of first paragraph in two (presently too long)

Monotherapy should not be hyphenated.

In the last sentence of the Intro, please restate all three strategies, not simply one.

Results:

Please use active voice and add an introductory sentence to the first paragraph. Omit repeat of “in Benin” in the 2nd sentence. Please state the numbers in each study arm in the first paragraph.

“Of those eligible, more than 95% started TPT”—please state the eligibility criteria.

Please add subheading to the Results.

Discussion:

HHCs not HHC’s

Please avoid jargon, e.g. stock-out

Please discuss the study limitations in the Discussion section.

We also request that you address the following PLOS Medicine requirements:

* PLOS Medicine requires that the de-identified data underlying the specific results in a published article be made available, without restrictions on access, in a public repository or as Supporting Information at the time of article publication, provided it is legal and ethical to do so. Please see the policy at

http://journals.plos.org/plosmedicine/s/data-availability

and FAQs at

http://journals.plos.org/plosmedicine/s/data-availability#loc-faqs-for-data-policy

* The Data Availability Statement (DAS) requires revision. For each data source used in your study:

a) If the data are freely or publicly available, note this and state the location of the data: within the paper, in Supporting Information files, or in a public repository, and include the DOI or accession number.

* For studies in which a novel model is central to the manuscript's findings, if this is the case here, authors are responsible for providing the source code needed to replicate the study's findings in a repository (such as GitHub, SourceForge or Bitbucket) or a cloud computing service (such as Code Ocean). Protection of authors’ intellectual property will not be cause for exception. Please explain in the manuscript’s Data Availability Statement how readers can access the shared code if new code was generated.

Comments from Reviewers:

Reviewer #1: Thanks to the authors for addressing all my previous comments well. I don't have any further issues to raise.

Reviewer #2: Thanks to the authors for their work in addressing all feedback! I think the authors have adequately addressed my comments, and have explained well where they disagreed. I find the findings of the paper are much easier to follow now and clarity has greatly improved. I have no further comments.

Reviewer #3: Thank you for addressing my comments - I am satisfied with your responses.

Congratulations on the study!

[LINK]

---

## [Editor Report · Decision Letter 3]

Dear Dr Menzies, 

On behalf of my colleagues and the Academic Editor, Amitabh Suthar, I am pleased to inform you that we have agreed to publish your manuscript "Rapid molecular testing or chest-X-ray or tuberculin skin testing for household contact assessment of tuberculosis infection: A cluster-randomized trial." (PMEDICINE-D-24-03440R3) in PLOS Medicine.

PRESS

Sincerely, 

Alison Farrell, Ph.D. 

Senior Editor 

PLOS Medicine